# Exogenously-Sourced Ethylene Modulates Defense Mechanisms and Promotes Tolerance to Zinc Stress in Mustard (*Brassica juncea* L.)

**DOI:** 10.3390/plants8120540

**Published:** 2019-11-25

**Authors:** M. Iqbal R. Khan, Badar Jahan, Mohamed F Alajmi, Md Tabish Rehman, Nafees A. Khan

**Affiliations:** 1Plant Systems Biology Laboratory, Department of Botany, School of Chemical and Life Sciences, Jamia Hamdard, New Delhi 110065, India; 2Plant Physiology and Biochemistry Laboratory, Department of Botany, Aligarh Muslim University, Aligarh 202002, India; naziabadar.2014@gmail.com (B.J.); naf9.amu@gmail.com (N.A.K.); 3Department of Pharmacognosy, King Saud University, Riyadh 11362, Kingdom of Saudi Arabia; malajmii@ksu.edu.sa (M.F.A.); m.tabish.rehman@gmail.com (M.T.R.)

**Keywords:** antioxidant system, ethylene, glyoxalase system, photosynthesis, proline metabolism, zinc

## Abstract

Heavy metal (HM) contamination of agricultural soil is primarily related to anthropogenic perturbations. Exposure to high concentration of HMs causes toxicity and undesirable effects in plants. In this study, the significance of ethylene was studied in response of mustard (*Brassica juncea*) to a high level (200 mg kg^−1^ soil) of zinc (Zn) exposure. Plants with high Zn showed inhibited photosynthesis and growth with the increase in oxidative stress. Application of ethylene (as ethephon) to Zn-grown plants restored photosynthesis and growth by inhibiting oxidative stress through increased antioxidant activity, the proline metabolism glyoxalase system, and nutrient homoeostasis. The results suggested that ethylene played a role in modulating defense mechanisms for tolerance of plants to Zn stress.

## 1. Introduction

The addition of heavy metals (HMs) to agricultural soil has become a leading problem worldwide due to a wide range of unrestricted and continuous anthropogenic activities, especially in developing countries. In recent times, the increasing concentration of HMs has invited attention in global research due to their non-biodegradability and their high accumulation in living things through the food web [1]. Heavy metals that serve as micronutrients, such as copper (Cu), iron (Fe), manganese (Mn), cobalt (Co), nickel (Ni), and zinc (Zn) play an essential function in plant growth and development under optimal concentrations [2,3]. Zinc, included in this category, has essential roles in binding of protein, regulation of enzyme activity, transcriptional and translational regulation, and signal transduction in being a component and co-factor of several enzymes [4,5]. The deficiency of Zn results in necrotic spots, leaf chlorosis, nutrient imbalance, altered cell division, and reduced photosynthesis and growth [6,7]. The presence of 8.0–100 μg g^−1^ DW Zn has been suggested to assist normal growth and development of plants, but at elevated concentrations of 300 µg g^−1^ it becomes a toxic pollutant [8,9] and causes overproduction of reactive oxygen species (ROS) [10,11]. Under these conditions, it significantly induces cellular damages, redox imbalance, replacement of essential functional groups, and inhibition of photosynthetic and growth processes [12].

Plants have inherent capabilities to strive under such stressful environments by way of modifying diverse defense mechanisms to scavenge/regulate the excess ROS in plants. The antioxidant enzymes, such as superoxide dismutase (SOD), ascorbate peroxidase (APX), glutathione peroxidase (GPX), glutathione S-transferase (GST), glutathione reductase (GR), monodehydroascorbate reductase (MDHAR), and dehydroascorbate reductase (DHAR), are activated for survival [13,14]. The co-substrates of ascorbate-glutathione (AsA-GSH) cycle, GSH and AsA, serve as non-enzymatic antioxidants for direct scavenging of ROS [15]. Under stress condition, increased production of a highly reactive substance, α-ketoaldehydes called methylglyoxals (MG), can damage cellular ultrastructure, causing inactivation of proteins and even cell death [16]. Consequently, efficient MG detoxification has become a necessary biochemical indicator for stress tolerance, where reduced glutathione (GSH)-dependent glyoxalase pathway efficiently performs MG detoxification via glyoxalase I (Gly I) and glyoxalase II (Gly II) enzymes [16]. Another potential mechanism for tolerance of plants to HM stress is proline accumulation [17]. Proline potentiality detoxifies excess ROS, maintains cellular osmotic environment, protects biological membranes, and stabilizes enzymes/proteins under stress conditions [18]. Thus, proline metabolism may also be considered as a strategy to increase tolerance of plants to HMs and to protect photosynthesis. 

Ethylene is a simple gaseous plant hormone that interacts with nutrient uptake and potentially influences many developmental processes of plants, including photosynthesis, under optimal and stressful conditions [19,20]. Iqbal et al. [21] showed that supplementation of ethephon resulted in increased activity of nitrate reductase and ATP-sulfurylase, which accounted for enhanced assimilation of nitrogen (N) and sulfur (S) resulting in increased antioxidant activity and photosynthetic responses in mustard plants. In wheat plants under heat stress [22] and cadmium (Cd) stress [23], ethylene regulates proline production. In spite of the fact that ethylene has now been recognized as an important modulator of photosynthetic process under optimal and stressful environments, its involvement in regulation of plant tolerance to elevated Zn levels through coordination of antioxidant and glyoxalase enzyme systems and proline metabolism has not been clearly demonstrated. It is postulated that the measures that could increase N-use efficiency in plants may result in greater investment of N in cellular metabolites and induce mechanisms that could protect photosynthesis and plant dry mass, and confer tolerance to Zn stress, through involvement of enhanced activity of antioxidant and glyoxalase systems and proline biosynthesis. The exogenously applied ethylene could help in achieving these processes and in alleviation of the adverse effects of Zn-induced stress.

## 2. Material and Methods

### 2.1. Plant Material, Growth Conditions, and Treatments

Healthy seeds of mustard (*Brassica juncea* L. Czern & Coss. cv. Varuna) were sterilized using 0.01 g L^−1^ mercuric chloride solution and repeated washings with double distilled water. The seeds were sown in earthen pots filled with soil having peat and compost, 4:1 (*v/v*), and mixed with sand, 3:1 (*v/v*). In each pot, five healthy plants were maintained after seedling establishment. The experiments were conducted at the Department of Botany, Aligarh Muslim University, Aligarh, India in the naturally illuminated green house. The plants grown with 200 mg Zn kg^−1^ soil (considered on the basis of our earlier research) [24] were treated with 200 μL L^−1^ ethephon at 20 days after sowing (DAS). The source for Zn was ZnSO_4_. In addition, a control group of plants and plants treated with 200 μL L^−1^ ethephon alone were also maintained. The application of 200 μL L^−1^ ethephon was done with a hand sprayer along with 0.5% teepol as surfactant. To maintain the effects of ethephon releasing ethylene, a high soil phosphorus (P) status was maintained as described earlier as Khan and Khan [24]. The treatments were arranged in a completely randomized block design and five replicates for each treatment were maintained (*n* = 3). All measurements were done at 30 DAS to record different physiological, biochemical, and growth attributes.

### 2.2. Measurements of Photosynthetic Traits and Plant Dry Mass Accumulation 

Gas exchange parameters (stomatal conductance (gs), intercellular CO_2_ concentration (Ci), and net photosynthesis (Pn) were measured in the fully expanded uppermost leaves of plants in each treatment using infrared gas analyzer (CID-340, Photosynthesis System, Bio-Science, Washington, USA). The measurements were done between 11:00 and 12:00 at light saturating intensity and at 370 ± 5 μmol mol^−1^ atmospheric CO_2_ concentration.

Chlorophyll content was measured with the help of a SPAD chlorophyll meter (SPAD 502 DL PLUS, Spectrum Technologies, Aurora, IL, USA).

Activity of ribulose-1,5-bisphosphate carboxylase (Rubisco) (EC 4.1.1.39) was determined spectrophotometrically by the adopting the method of Usuda [25] by monitoring nicotinamide adenine dinucleotide (NADH) oxidation at 30 °C at 340 nm during the conversion of 3-phosphoglycerate to glycerol 3-phosphate after the addition of enzyme extract to the assay medium. Further detail is given by Khan and Khan [24].

By using a Junior-PAM chlorophyll fluorometer (Heinz Walz, Effeltrich, Germany) all the chlorophyll fluorescence parameters were studied as described earlier by Khan and Khan [24]. Calculations were done according to Khan and Khan [24] and Krall and Edwards [26].

Dry mass of plants was recorded after drying the sample in a hot air oven at 80 °C till constant weight. Leaf area was measured using a leaf area meter (LA211, Systronics, New Delhi, India).

### 2.3. Determination of Oxidative Stress Markers

#### 2.3.1. Lipid Peroxidation

Lipid peroxidation in leaves was determined by estimating the content of thiobarbituric acid reactive substances (TBARS) as described by Dhindsa et al. [27]. The content of TBARS was calculated using the extinction coefficient (155 mM^−1^ cm^−1^). The details have been given in our earlier report as Khan et al. [22].

#### 2.3.2. Electrolyte leakage

For measuring electrolyte leakage, samples were thoroughly washed with sterile water, weighed, and then kept in closed vials with 10 mL of deionized water where they were incubated at 25 °C for 6 h using a shaker. Then, electrical conductivity (EC) was determined (C1). Samples were then again kept at 90 °C for 2 h and EC was recorded after attaining equilibrium at 25 °C (C2).

#### 2.3.3. Methylglyoxal Content

Adopting the method of Wild et al. [28], methylglyoxal content was measured. Leaf samples were homogenized by using 5% perchloric acid followed by centrifugation at 11,000× *g*. A saturated solution of Na_2_CO_3_ was used to neutralize the supernatant and further mixed with Na-P and N-acetyl-L-cysteine. The product N-α-acetyl-S-(1-hydroxy-2-oxoprop-1-yl) cysteine formation was measured by spectrophotometer at 288 nm. A standard curve of MG was prepared and expressed in μmol g^−1^ FW.

#### 2.3.4. Lipoxygenase Activity

Lipoxygenase (LOX) (EC 1.13.11.12) activity was estimated following the method of Doderer et al. [29] by monitoring the increase in absorbance at 234 nm using linoleic acid as a substrate. The LOX activity was calculated using 25 mM^−1^ cm^−1^ as an extinction coefficient.

### 2.4. Measurement of Ascorbate and Glutathione Content

#### 2.4.1. Ascorbate Content

Ascorbate (AsA) content was determined following the method of Law et al. [30] with some modifications. The details have been given in our earlier report as Anjum et al. [31]. A standard curve in the range of 10–100 nmol of ascorbic acid was used for calibration. Values in both cases were corrected for the absorbance eliminating the supernatant in the blank prepared separately for AsA. 

#### 2.4.2. Reduced Glutathione Content

GSH was assayed by an enzymic recycling procedure, as detailed by Griffith [32], in which it was sequentially oxidized by 5,5-dithiobis-2-nitrobenzoic acid (DTNB) and reduced by NADPH in the presence of GR, as described earlier [24,33].

### 2.5. Extraction and Determination of Antioxidant Enzymes 

Fresh leaf tissue (200 mg) was homogenized with an extraction buffer containing 0.05% (*v/v*) Triton X-100 and 1% (*w/v*) PVP in potassium-phosphate buffer (100 mM, pH 7.0) using chilled mortar and pestle. At 4 °C, the homogenate was centrifuged at 15,000× *g* for 20 min. The supernatant obtained after centrifugation was used to assay superoxide dismutase (SOD) (EC 1.15.1.1), GSH reductase (GR) (EC 1.6.4.2), and GSH peroxidase (GPX) (EC 1.11.1.9) enzymes, and for the assay of ascorbate peroxidase (APX) (EC 1.11.1.11). Here, 2.0 mM ascorbate was supplemented with extraction buffer. Protein was estimated according to Bradford [34] using bovine serum albumin as standard.

#### 2.5.1. SOD

Activity of SOD was determined by monitoring the inhibition of photochemical reduction of nitro blue tetrazolium (NBT), according to the methods of Beyer and Fridovich [35] and Giannopolitis and Ries [36]. The details have been given in our earlier report as Khan and Khan [24].

#### 2.5.2. APX

Activity of APX was determined by the method of Nakano and Asada [37] by recording the decrease in absorbance of ascorbate at 290 nm. The assay mixture contained phosphate buffer (50 mM, pH 7.0), 0.1 mM EDTA, 0.5 mM ascorbate, 0.1 mM H_2_O_2_, and the enzyme extract. APX activity was calculated by using the extinction coefficient 2.8 mM^−1^ cm^−1^. One unit of the enzyme is the amount necessary to decompose 1 μmol of substrate per min at 25 °C.

#### 2.5.3. GR

Activity of GR was determined adopting the method of Foyer and Halliwell [38] by monitoring the GSH-dependent oxidation of NADPH. The details have been given in our earlier report as Khan and Khan [24]. 

#### 2.5.4. GPX

Activity of GPX was determined by adopting the method of Hasanuzzaman et al. [39]. The details have been given in our earlier report as Khan et al. [24]. 

#### 2.5.5. GST

Glutathione S-transferase (GST) (EC 2.5.1.18) was determined spectrophotometrically by the method Booth et al. [40] with some modifications [41]. The reaction mixture contained 100 mM Tris–HCl buffer (pH 6.5), 1.5 mM GSH, 1 mM 1-chloro-2,4-dinitrobenzene (CDNB), and enzyme solution in a final volume of 0.7 mL. The enzyme reaction was initiated by the addition of CDNB and the increase in absorbance was measured at 340 nm for 1 min. The activity was calculated using the extinction coefficient of 9.6 mM^−1^ cm^−1^.

#### 2.5.6. MDHAR

Monodehydroascorbate reductase (MDHAR) (EC 1.6.5.4) activity was assayed by the method of Hossain et al. [42]. The reaction mixture contained 50 mM Tris–HCl buffer (pH 7.5), 0.2 mM NADPH, 2.5 mMAsA, and 0.5 units of AO and enzyme solution in a final volume of 0.7 mL. The reaction was started by the addition of AO. The activity was calculated from the change in ascorbate at 340 nm for 1 min using an extinction coefficient of 6.2 mM^−1^ cm^−1^. 

#### 2.5.7. DHAR

Dehydroascorbate reductase (DHAR) (EC 1.8.5.1) activity was assayed by the method of Nakano and Asada [37]. The reaction buffer contained 50 mM K-phosphate buffer (pH 7.0), 2.5 mM GSH, and 0.1 mM DHA. The reaction was started by adding the sample solution to the reaction buffer solution. The activity was calculated from the change in absorbance at 265 nm for 1 min using an extinction coefficient of 14 mM^−1^ cm^−1^.

### 2.6. Extraction and Assay of Glyoxalase Systems’ Enzymes

#### 2.6.1. Gly I Activity

Glyoxalase I (Gly I) (EC 4.4.1.5) assay was determined by the method of Hasanuzzaman et al. [43]. The assay mixture contained 100 mM K-phosphate buffer (pH 7.0), 15 mM MgSO_4_, 1.7 mM GSH, and 3.5 mM MG in a final volume of 700 µL. The reaction was started by the addition of MG and the increase in absorbance was recorded at 240 nm for 1 min. The activity was calculated using the extinction coefficient of 3.37 mM^−1^ cm^−1^.

#### 2.6.2. Gly II Activity

Glyoxalase II (Gly II) (EC 3.1.2.6) activity was measured using the method of Principato et al. [44] by monitoring the formation of GSH at 412 nm for 1 min. The reaction mixture contained 100 Mm Tris–HCl buffers (pH 7.2), 0.2 mM DTNB and 1 mM S-D-lactoylglutathione (SLG) in a final volume of 1 mL. The reaction was started by the addition of SLG and the activity was calculated using the extinction coefficient of 13.6 mM^−1^ cm^−1^.

### 2.7. Determination of Nutrient Content

The determination of mineral nutrients (nitrogen, N; phosphorous, P; potassium, K; and calcium, Ca) was done in acid-peroxide digested oven-dried leaf sample. K, and Ca were measured using flame photometer (Khera-391: Khera Instruments, New Delhi, India), whereas N and P were determined by using the methods of Lindner [45] and Fiske and Subba Row [46], respectively.

### 2.8. Determination of Proline Content and Activity of Proline Metabolizing Enzymes

#### 2.8.1. Proline Content

Proline content was determined by adopting the ninhydrin method of Bates et al. [47]. Here, 300 mg fresh leaf samples were homogenized in 3% sulphosalicylic acid (3 mL). After this, samples were homogenate filtrated and reacted with acid ninhydrin and glacial acetic acid (1 mL each) for 1 h followed by water bath at 100 °C. The reaction mixture was extracted with toluene and the absorbance was measured at 520 nm. A standard was also prepared using L-proline.

To determine the activity of γ-glutamyl kinase (GK) (EC 2.7.2.11) and proline oxidase (PROX) (EC 1.5.99.8), enzyme extract was prepared by homogenizing 500 mg leaf sample in 0.1 M Tris–HCl buffer (pH 7.5) at 4 °C. The homogenate was centrifuged at 30,000× *g* for 30 min and the supernatant was used as the crude extract enzyme preparation for P5CS activity while the pellet was collected and used as extract for the assay of GK and proline oxidase.

#### 2.8.2. GK Activity

Activity of GK was assayed by the method of Hayzer and Leisinger [48] with slight modification. The frozen sample was suspended in 10 mL of 0.1 M Tris–HCl buffer containing 1 mM 1,4-dithiothreitol (DTT) to rupture the cell and centrifuged at 30,000× *g* for 30 min. The other detail has been given in our earlier report as Khan et al. [22]. Activity of GK was expressed in U mg^−1^ protein. One unit of the enzyme activity is defined as μg of glutamyl hydroxamate min^−1^ mg^−1^ protein. Glutamyl hydroxamate was used as standard.

#### 2.8.3. POX Activity

Activity of POX was determined adopting the method of Huang and Cavalieri [49] with slight modification. The pellet was mixed with 1 mL Tricine and KOH buffer (pH 7.5) containing 6 M sucrose. This extract was used for the enzyme assay. The other detail has been given in our earlier report as Khan et al. [22]. Proline oxidase activity was expressed in U mg^−1^ protein. One unit of the enzyme activity is defined as mM DCPIP reduced min^−1^ mg^−1^ protein.

### 2.9. Water Potential and Osmotic Potential

Leaf water potential was measured on the second leaf from the top (fully expanded young leaf) of the plant by using the water potential system (Psypro, WESCOR, UT, USA). The leaf used for water potential measurement was frozen in liquid N_2_ in sealed polythene bags which were thawed, and cell sap was extracted with the help of a disposable syringe. The extracted sap was used for the determination of osmotic potential using a vapor pressure osmometer (5520, WESCOR, UT, USA). 

### 2.10. Statistical Analysis

Data were analyzed statistically, and standard error was calculated. Analysis of variance was performed on the data using SPSS (ver. 17.0 Inc.) to determine the significance at *p* < 0.05. Least significant difference (LSD) was calculated for the significant data to identify difference in the mean of the treatment; data are presented as mean ± SE (*n* = 3).

## 3. Results

### 3.1. Ethephon Reverses Effects of Zn Stress on Photosynthetic and Growth Attributes

Photosynthetic attributes were reduced in plants treated with Zn compared to the control plants. The adverse effect of Zn stress on photosynthetic parameters was reversed with ethephon application. Ethephon application to plants without Zn treatment enhanced net photosynthesis by 27.2%, stomatal conductance by 20.5%, intracellular CO_2_ concentration by 17.2%, Rubisco content by 28.8%, and chlorophyll content by 25.7%, as compared to the control plants. In plants treated with Zn, ethephon supplementation restricted the adverse effects of Zn and the decrease in net photosynthesis was reduced to 18.1%, stomatal conductance to 13.1%, intracellular CO_2_ concentration to 15.4%, Rubisco content to 16.4%, and chlorophyll content to 16%, in comparison to control (Table 1).

Plants grown with Zn stress exhibited a decrease in maximum PSII efficiency, intrinsic PSII efficiency, actual PSII efficiency, photochemical quenching, and electron transport rate by 26%, 23%, 29%, 24%, and 25% in comparison with the control plants. However, non-photochemical quenching increased with Zn by 59.2% as compared to the control. Ethephon application to plants without Zn stress improved all the above traits compared with control (Table 1).

Growth of plants was reduced with Zn when compared to the control plants. Ethephon application to plants without Zn treatment enhanced plant dry mass and leaf area compared to the control plants. In plants treated with Zn, ethephon supplementation restricted the adverse effects of Zn and the decrease in plant dry mass was limited to 15% as compared to control plants. Zn reduced leaf area by 30.3% compared to control (Table 1). Ethephon application to plants without Zn treatment enhanced plant dry mass and leaf area compared to the control plants. The increases in plant dry mass and leaf area were 23% and 27%, respectively, compared to the control (Table 1).

### 3.2. Influence of Ethephon on Water Relations under Zn Stress

Application of ethephon to Zn-stressed plants increased water potential by 35% and osmotic potential by 47% as compared to the control plants. On the other hand, application of Zn alone resulted in the increase of water potential by 141% and osmotic potential by 71.4% as compared to the control plants (Figure 1).

### 3.3. Ethephon Reduces Zn-Induced Oxidative Stress

Plants grown with Zn showed higher oxidative stress and exhibited increased oxidative stress markers, i.e., content of TBARS and electrolyte leakage, MG content, and LOX activity (Table 2). Treatment of plants with Zn enhanced TBARS content, MG content, and LOX activity by about 95%, 61% and 51%, respectively, whereas electrolyte leakage increased by about 2.2 times in plants compared to the control. For the appraisal of the influence of ethylene in reducing Zn-induced oxidative stress, we analyzed TBARS content and electrolyte leakage after application of ethylene to these plants. Application of ethylene proved effective in lowering oxidative stress under the metal stress. Application of ethylene reduced TBARS content, MG content, and LOX activity by 28%, 13% and 12%, respectively whereas electrolyte leakage was reduced by 47% in Zn-treated plants compared to Zn-treated plants (Table 2).

### 3.4. Application of Ethephon Enhanced Antioxidant Systems under Zn Stress

The activities of enzymatic and non-enzymatic antioxidant systems showed modulation with ethylene treatment in both stress and non-stress conditions. Zn stress decreased AsA content as compared to the control plants. On the other hand, ethephon supplementation of Zn-exposed plants had significantly higher AsA content as compared to Zn-treated plants. Under Zn stress, GSH content was increased by 22% as compared to control plants. Application of ethephon to Zn-exposed plants had significantly higher GSH content by 65% as compared to the control plants (Figure 2).

Zinc treatment resulted in increased activity of SOD, APX, GR, and GPX by 25%, 38%, 45%, and 35%, respectively compared to the control plants. In non-stressed control plants, application of ethephon increased activity of SOD APX, GR, and GPX by 50%, 128%, 105%, and 124%, respectively, over the control. However, application of ethephon to Zn-treated plants resulted in increased activity of SOD, APX, GR, and GPX by 71%, 176%, 145%, and 165%, respectively, compared to the control (Figure 3).

Zinc treatments also increased the activity of MDHAR, DHAR, and GST by 45%, 26%, and 67%, respectively, as compared to the control plants. Ethephon application to plants grown without Zn resulted in increased activity of MDHAR by 97%, DHAR by 53%, and GST by 219%, in comparison to control plants; in presence of Zn, ethephon application resulted in 170%, 76%, and 261% higher activity of MDHAR, DHAR, and GST, respectively, in comparison to the control plants (Figure 4).

### 3.5. Ethephon Application Enhances Glyoxalase System

Gly I was increased with Zn stress by 73% over the control plants. Application of ethephon to Zn-treated plant significantly enhanced the activity of Gly I by 94% compared to the control plants (Figure 5). The increase in Gly II with Zn stress was 110% in comparison to the control plants. However, application of ethephon to Zn-treated plants enhanced the activity of Gly II by 160% as compared to the control plants (Figure 5).

### 3.6. Ethephon Increases Proline Metabolism under Zn Stress

In order to assess the role of proline metabolism in Zn stress tolerance, assessment of proline accumulation and activity of proline metabolizing enzymes (GK and PROX) was done. Activity of GK increased in Zn-stressed plants and also with ethephon plus Zn stress treatments. Application of ethephon increased GK activity by 170% in Zn-stressed plants compared to the control. On the other hand, activity of PROX was reduced under no-stress and Zn-stressed plants with ethephon treatment (Figure 6). 

Proline accumulation increased upon ethephon application, as well as with the Zn treatments. Zinc stress induced proline biosynthesis and increased proline content by 44% in comparison to the control plants. Proline accumulation under Zn stress was further increased with ethephon application. Maximum proline accumulation resulted from ethephon application under Zn stress compared with the control (Figure 6).

### 3.7. Ethephon Supplementation Maintained Nutrient Contents under Zn Stress

Treatment of ethephon alone significantly increased the content of nutrients but Zn stress decreased N, P, K, and S content in comparison to the control plants. Zn treatment decreased N by 29%, P by 23%, K by 31%, and S by 34%, as compared to control. However, application of ethephon on Zn-grown plants completely alleviated the Zn effects and increased the nutrient contents significantly in comparison to stressed plants (Figure 7).

## 4. Discussion

Zinc stress inhibited photosynthesis and plant growth in mustard plants. However, ethephon supplementation not only stimulated photosynthesis and growth under non-stress conditions but also under Zn stress. Plants show concentration-dependent requirement of Zn for optimal plant metabolism; however, excess Zn availability inhibits plant growth and development [24]. It has been reported that HMs induce lipid peroxidation in photosynthetic membranes, distort chloroplast ultrastructure, degrade photosynthetic pigments, inhibit PSII activity and the electron transport chain, and decrease both carboxylation efficiency of Rubisco and net photosynthesis [50,51]. Additionally, plants’ fitness to the environment is estimated through chlorophyll fluorescence observations [52]. In the present study, the reduction in photochemical efficiency under Zn stress (Table 1) indicated the interruption in photochemical reactions that blocked electron transport system and the performance of PSII resultantly increased NPQ under Zn stress (Table 1). The results of previous study of Sirhindi et al. [53] have also reported that NPQ increases under Ni stress that was due to repressed photochemistry; this was considered as the mechanism to balance the excess absorbed light energy preventing photo inhibition.

It has been reported that ethylene potentially plays a crucial role in the adaptation of plants to HM stress [54]. Ethephon is used commercially to produce ethylene in crop plants and the ethylene released from ethephon affects several cellular, developmental, photosynthetic, and stress response processes [55]. The present study showed that exogenous ethephon supply decreased the toxic effects of Zn on photosynthetic machinery (Table 1) and enhanced the activity of PSII and gas exchange parameters. Our results are consistent with the finding of Asgher et al. [56], who reported that ethephon supplementation induced photosynthesis under Cr stress. 

In the present study, Zn-stressed plants were at substantially increased oxidative stress and MG levels exhibiting increased LOX activity. These observations in mustard plants are in agreement with the finding of Molassiotis et al. [57], who found increased LOX activity which was the result of increased production of ROS due to higher oxidative stress in *Malus domestica* shoot tips under boron stress. Our results showed a possible regulatory role of exogenously applied ethylene as ethephon on ROS and MG metabolism in Zn stress tolerance in mustard plants. Indeed, ethylene acts as a vital signaling molecule for reduction of ROS [19,58]. Earlier, it has been shown that coordination and regulation of the antioxidant systems and glyoxalase systems are indispensable to attain significant tolerance against oxidative stress [59,60]. We have found that the activity of antioxidant enzymes and glyoxalase system simultaneously work to reduce excess ROS load and MG detoxification.

Antioxidant enzyme systems are part of an imperative defense system of plants against ROS caused by HMs [61]. Application of ethephon substantially induced the activity of SOD, APX, GR, GST, GPX, MDHAR, and DHAR in plants under Zn stress. Further, ethylene application enhanced the activities of Gly I and Gly II, particularly in response to Zn stress. Recently, Hassanuzzaman et al. [62] showed that the activities of Gly I and Gly II in *Oryza sativa* increased under Ni stress and Si application. Studies have shown that ethephon treatment influenced the activity of antioxidant enzymes and photosynthetic attributes, but the information on the response of plants to ethephon under Zn stress is scanty. In the present study, the effort was made to understand the role of ethylene in regulation of three major defense mechanisms (antioxidant system, glyoxalase system, and proline metabolism) along with nutrient homeostasis under high level of Zn in mustard plants. Ethylene significantly induced the defense mechanisms by up-regulating the key enzymes in the antioxidant system (SOD, APX, GR, GPX, GST), glyoxalase system (Gly I and Gly II) and proline metabolism (GK and PROX), which resulted in protection of growth and development of plants through maintaining water relations and increasing PS II activity and dry mass production. 

In addition to the antioxidant and glyoxalase defense system, we also examined proline as a defense mechanism under Zn stress and to know how it was modulated with ethylene. Plants under Zn stress accumulated proline. These results are supported with the previous findings of Sirhindi et al. [53], who have shown that proline content increases in *Glycine max* under Ni toxicity. Alia et al. [63] showed that proline was involved in reducing the photodamage in the thylakoid membranes by scavenging ROS (singlet oxygen and superoxide radical anion). In the present study, ethylene-induced accumulation of proline helped in increasing osmotic potential, and thereby water potential, and protected PSII activity and photosynthesis by reducing ROS effects under Zn stress. The activity of chlorophyll fluorescence was reduced in Zn-stressed plants receiving ethylene that showed improved photosynthesis. Earlier, it has been shown that ethephon application increased the gas exchange parameters and PSII activity in mustard under HM stress [56,58]. Proline biosynthesis has been attributed in the alleviation of cytoplasmic acidosis and may maintain NADP^+^/NADPH ratio at values compatible with plant metabolism. Rapid production of proline under stress also provides recovery from stress-induced damages to mitochondrial oxidative phosphorylation and the generation of ATP [64]. However, there has been much disagreement regarding the mechanisms by which proline reduces HM-induced oxidative stress, regulates cellular functions such as osmotic adjustment, and enhances the plant’s water status [14]. In the present research, efforts were made to study the role of proline and its metabolizing enzymes in facilitating Zn detoxification and ameliorating stress in mustard plants. The increased proline accumulation after ethephon application resulted from induced GK activity and inhibited PROX activity. It has been reported that activities of GK and PROX play important roles in controlling the level of proline and environmental stress in plants [65,66]. The regulatory role of ethylene in proline metabolism and its enzymes to improve photosynthetic traits under Cd and heat stress has been shown [21,22].

HMs impact negatively on the environment and crop nutrition [67]. They disturb the uptake of essential nutrients such as N, P, K, and S which are required for normal growth of plants [68,69]. In the present study, we also focused on modulation of essential nutrients (N, P, K, and S) in mustard plants through application of ethephon under Zn stress. Ethylene has been associated with the regulation of physiological responses to nutrient homeostasis and stress tolerance responses. Efficient working of nutrient transporters and enzymes involved in nutrient assimilation enhanced nutrient uptake and this has a direct influence on the tolerance mechanisms including antioxidant system, glyoxalase system, and proline metabolism of plants under stress condition [70].

## 5. Conclusions and Future Prospects 

Conclusively, it may be said that Zn stress adversely impacted photosynthesis and growth of plants by inhibiting metabolic pathways. Ethylene supplementation inhibited ROS production through induced defense mechanisms of antioxidant activity, proline metabolism, glyoxalase system and nutrient homoeostasis. It protected photosynthetic machinery and promoted photosynthesis and growth under Zn stress. Thus, the use of ethylene (as ethephon) may bear a prominent role in alleviation of Zn stress in mustard plants by modulating the defense mechanisms.

## Figures and Tables

**Figure 1 plants-08-00540-f001:**
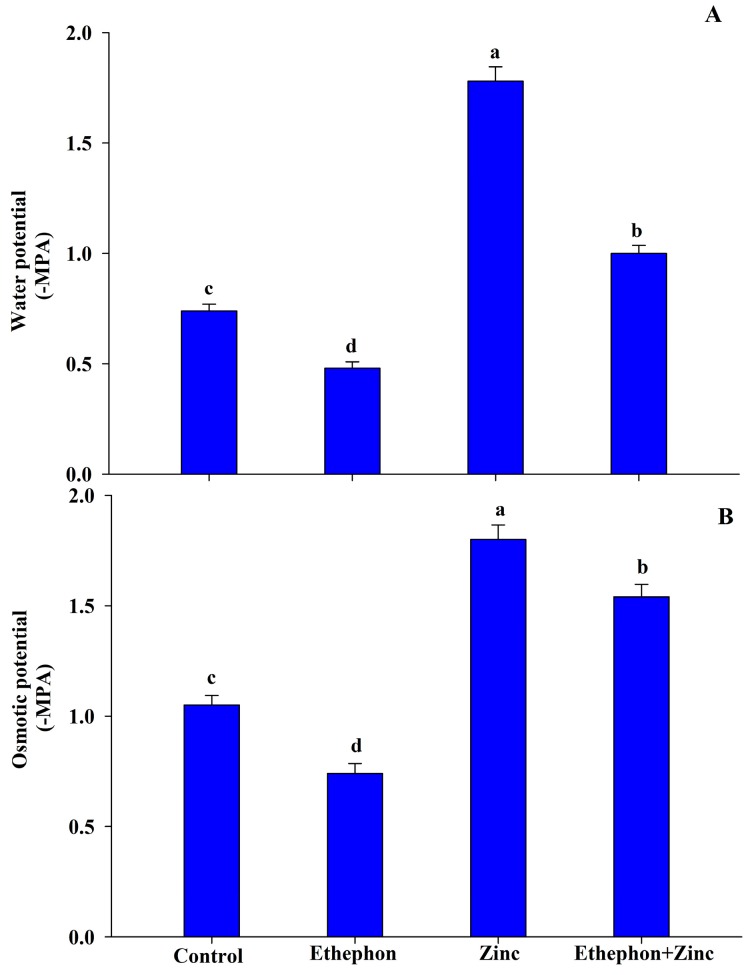
(**A**) Leaf water potential and (**B**) osmotic potential in the leaf of mustard (*Brassica juncea* L.) of Varuna cultivar at 30 DAS. Plants were grown with/without Zn stress and treated with 200 µL L^−1^ ethephon at 20 DAS. Data are presented as treatment mean ± SE (*n* = 3). Data followed by same letter are not significantly different by LSD test at *p* < 0.05.

**Figure 2 plants-08-00540-f002:**
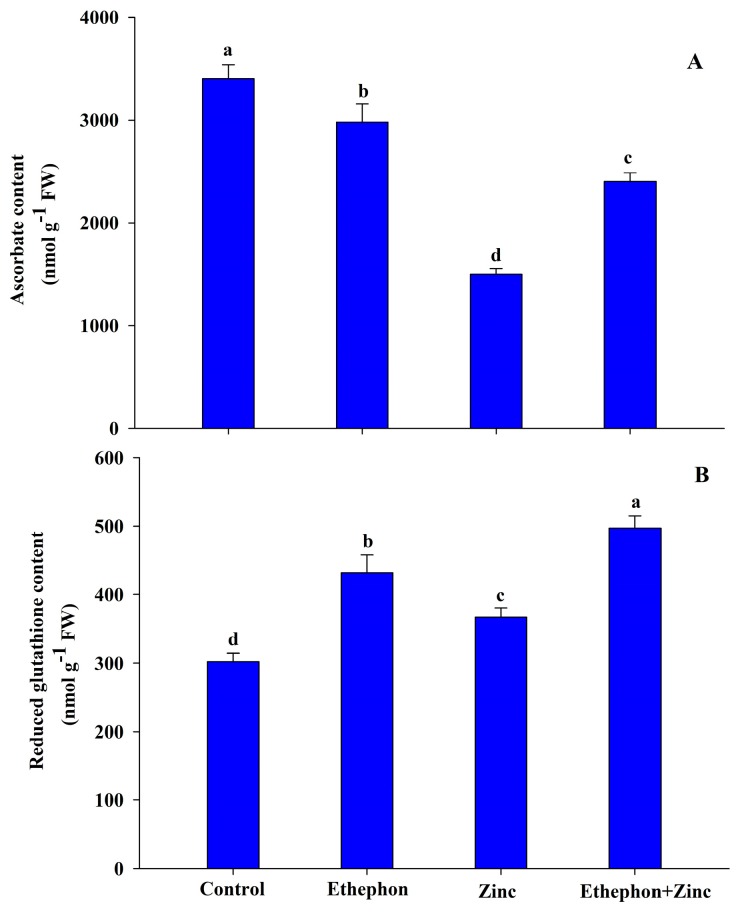
(**A**) Ascorbate content and (**B**) reduced glutathione content in the leaf of mustard (*Brassica juncea* L.) of Varuna cultivar at 30 DAS. Plants were grown with/without Zn stress and treated with 200 µL L^−1^ ethephon at 20 DAS. Data are presented as treatment mean ± SE (*n* = 3). Data followed by same letter are not significantly different by LSD test at *p* < 0.05.

**Figure 3 plants-08-00540-f003:**
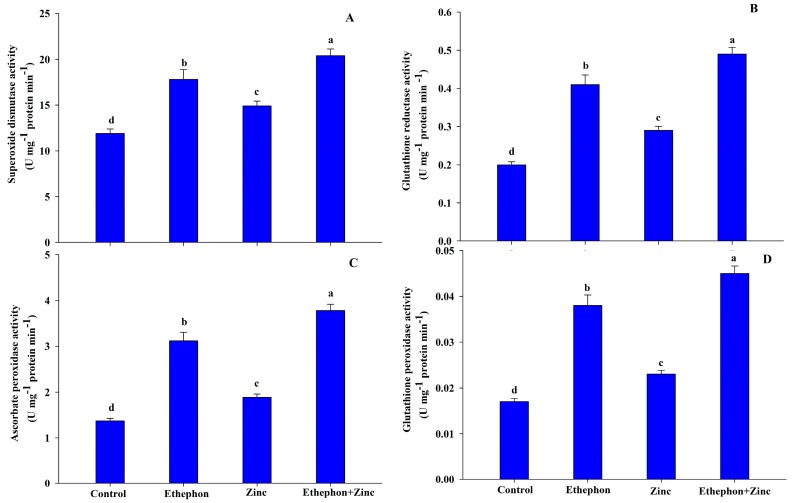
(**A**) Superoxide dismutase activity; (**B**) glutathione reductase activity; (**C**) ascorbate peroxidase activity; (**D)** and glutathione peroxidase activity in the leaf of mustard (*Brassica juncea* L.) of Varuna cultivar at 30 DAS. Plants were grown with/without Zn stress and treated with 200 µL L^−1^ ethephon at 20 DAS. Data are presented as treatment mean ± SE (*n* = 3). Data followed by same letter are not significantly different by LSD test at *p* < 0.05.

**Figure 4 plants-08-00540-f004:**
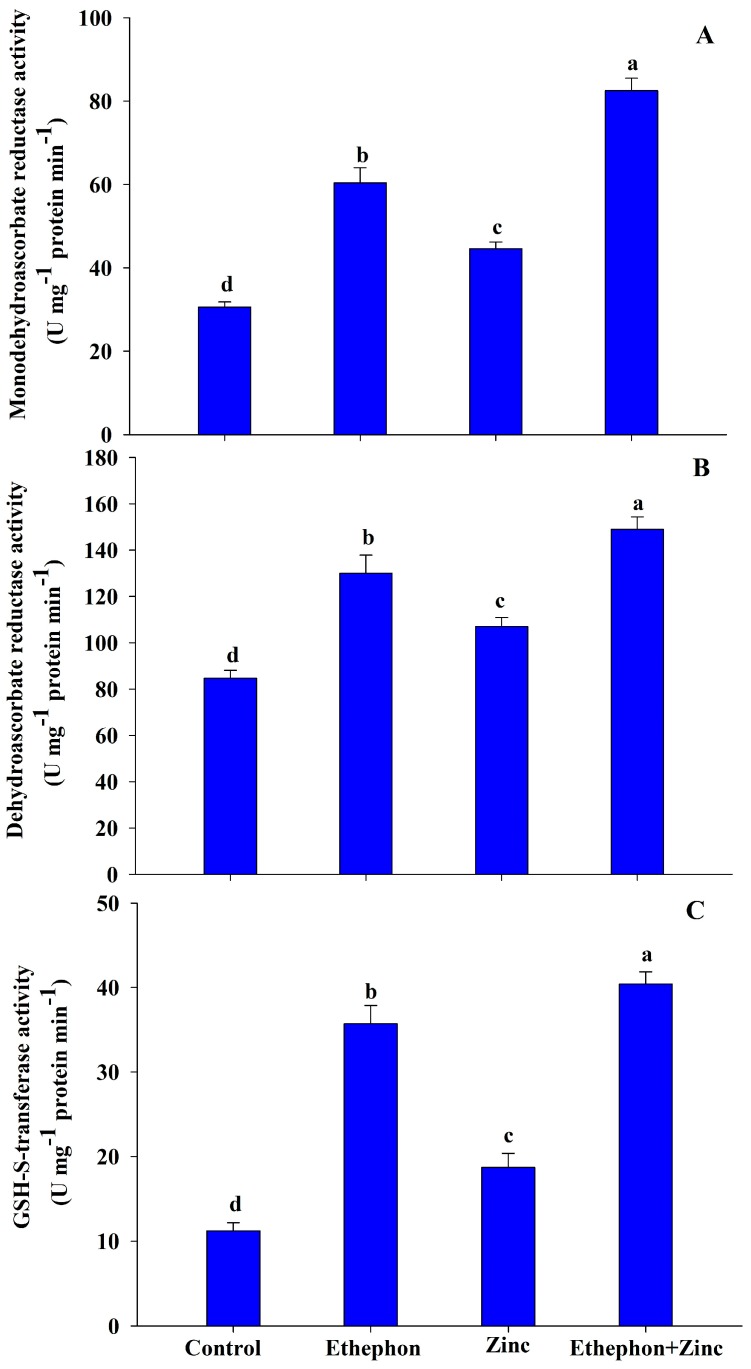
(**A**) Monodehydroascorbate activity; (**B**) dehydroascorbate activity; and (**C**) GSH-S-transferase activity in the leaf of mustard (*Brassica juncea* L.) of Varuna cultivar at 30 DAS. Plants were grown with/without Zn stress and treated with 200 µL L^−1^ ethephon at 20 DAS. Data are presented as treatment mean ± SE (*n* = 3). Data followed by same letter are not significantly different by LSD test at *p* < 0.05.

**Figure 5 plants-08-00540-f005:**
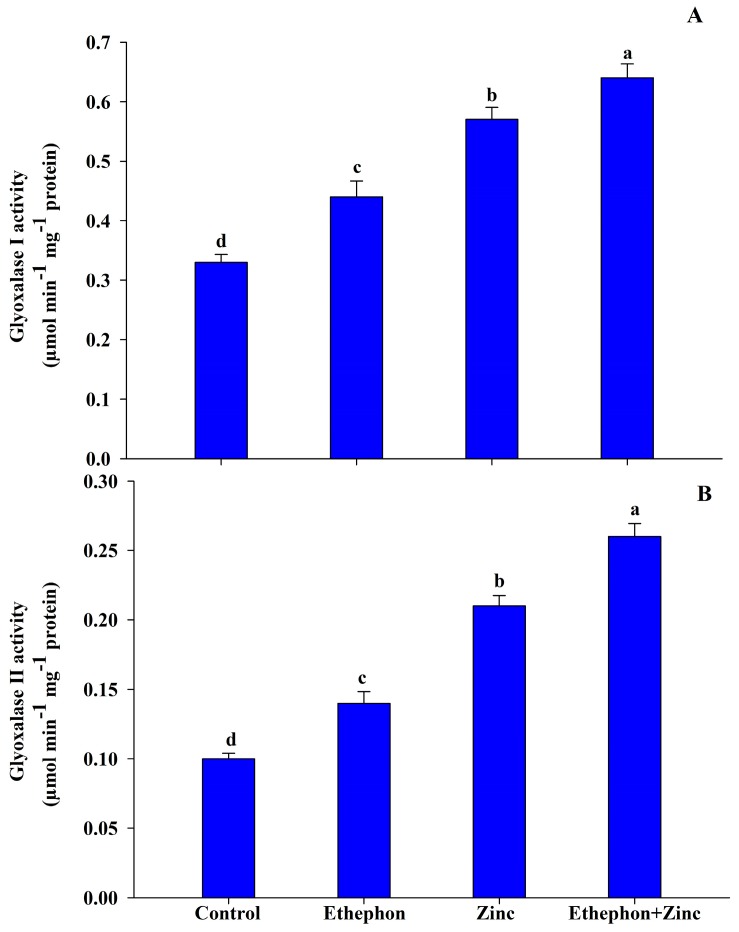
(**A**) Glyoxalase I activity and (**B**) glyoxalase II activity in the leaf of mustard (*Brassica juncea* L.) of Varuna cultivar at 30 DAS. Plants were grown with/without Zn stress and treated with 200 µL L^−1^ ethephon at 20 DAS. Data are presented as treatment mean ± SE (*n* = 3). Data followed by same letter are not significantly different by LSD test at *p* < 0.05.

**Figure 6 plants-08-00540-f006:**
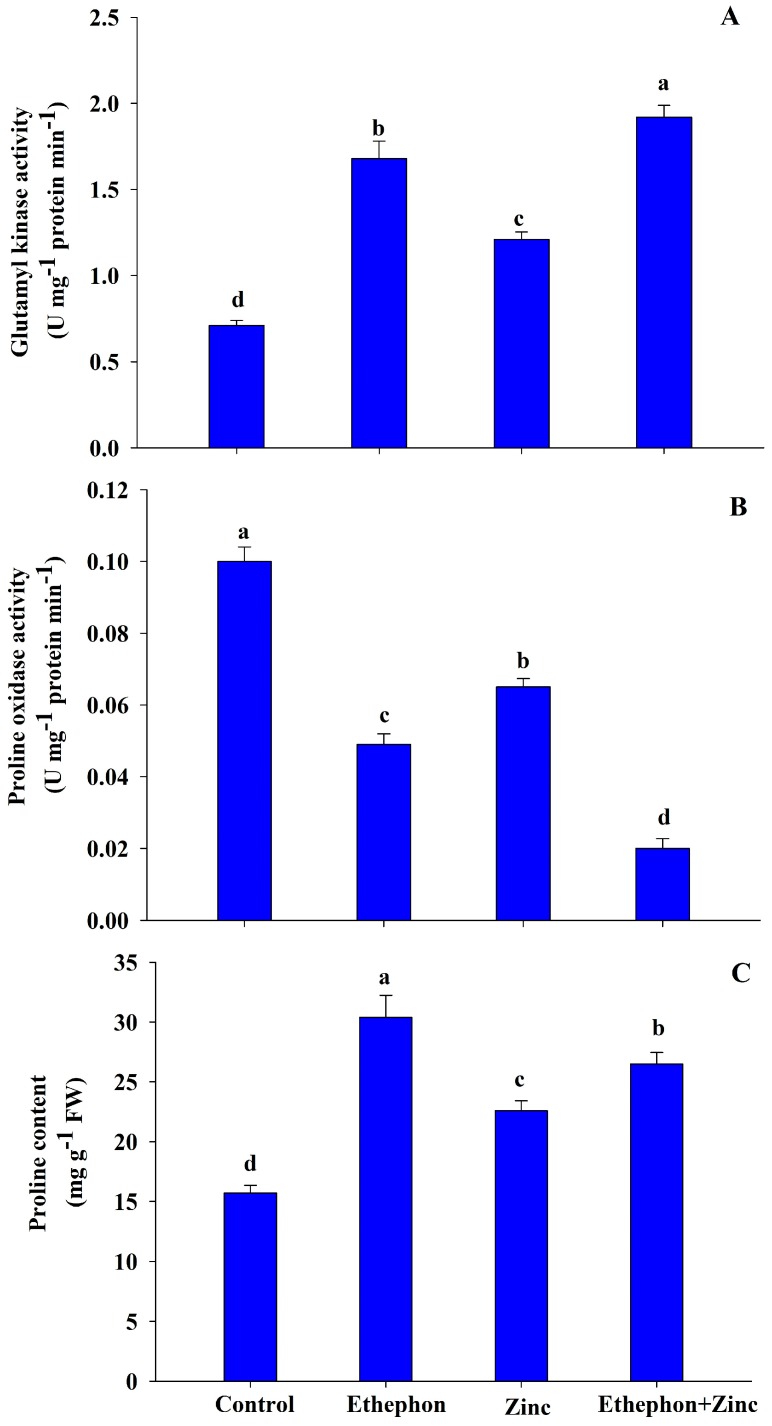
(**A**) Activity of glutamyl kinase; (**B**) proline oxidase; and (**C**) proline content in the leaf of mustard (*Brassica juncea* L.) of Varuna cultivar at 30 DAS. Plants were grown with/without Zn stress and treated with 200 µL L^−1^ ethephon at 20 DAS. Data are presented as treatment mean ± SE (*n* = 3). Data followed by same letter are not significantly different by LSD test at *p* < 0.05.

**Figure 7 plants-08-00540-f007:**
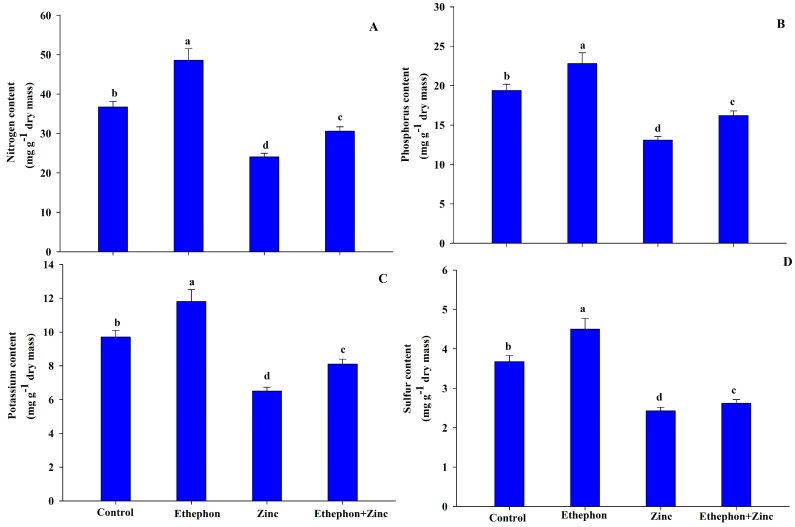
(**A**) Nitrogen content; (**B**) phosphorous content; (**C**) potassium content; and (**D**) sulfur content in the leaf of mustard (*Brassica juncea* L.) of Varuna cultivar at 30 DAS. Plants were grown with/without Zn stress and treated with 200 µL L^−1^ ethephon at 20 DAS. Data are presented as treatment mean ± SE (*n* = 3). Data followed by same letter are not significantly different by LSD test at *p* < 0.05.

**Table 1 plants-08-00540-t001:** Chlorophyll content, net photosynthesis rate, intracellular CO_2_ concentration, stomatal conductance, Rubisco activity, Φ PS II, Fv/Fm, Fv′/Fm′, qP, NPQ, ETR, plant dry mass, and leaf area in the leaf of mustard (*Brassica juncea* L.) cv. Varuna at 30 days after sowing (DAS). Plants were grown with/without zinc stress and treated with 200 µL L^−1^ ethephon at 20 DAS. Data are presented as treatment mean ± SE (*n* = 3). Data followed by same letter are not significantly different by least significant difference (LSD) test at *p* < 0.05.

	Control	Ethephon	Zinc	Ethephon + Zinc
Chlorophyll content (SPAD value)	31.4 ± 1.3 ^b^	39.5 ± 1.6 ^a^	21.3 ± 0.8 ^d^	26.4 ± 0.9 ^c^
Net photosynthesis (µmol CO_2_ m^−2^s^−1^)	19.8 ± 0.8 ^b^	25.2 ± 0.9 ^a^	11.2 ± 0.7 ^d^	16.2 ± 0.6 ^c^
Intracellular CO_2_ concentration (µmol CO_2_ mol^−1^)	272 ± 10.9 ^b^	319 ± 12.7 ^a^	187 ± 6.7 ^d^	230 ± 8.2 ^c^
Stomatal conductance (mmol CO_2_ m^−2^ s^−1^)	380 ± 10.7 ^b^	458 ± 12.3 ^a^	280 ± 10.1 ^d^	330 ± 11.8 ^c^
Rubisco activity (µmol CO_2_ m^−2^ s^−1^)	48.6 ± 1.9 ^b^	62.6 ± 2.2 ^a^	32.5 ± 1.17 ^d^	40.6 ± 1.5 ^c^
Actual PSII efficiency (Φ PS II)	0.62 ± 0.03 ^ab^	0.68 ± 0.04 ^a^	0.44 ± 0.02 ^c^	0.57 ± 0.02 ^b^
Maximum PSII efficiency (Fv/Fm)	0.77 ± 0.03 ^a^	0.84 ± 0.03 ^a^	0.57 ± 0.02 ^c^	0.67 ± 0.03 ^b^
Intrinsic PSII efficiency (Fv’/Fm’)	0.71 ± 0.03 ^b^	0.75 ± 0.03 ^a^	0.55 ± 0.02 ^c^	0.65 ± 0.02 ^b^
Photochemical quenching (qP)	0.83 ± 0.04 ^a^	0.87 ± 0.03 ^a^	0.63 ± 0.02 ^c^	0.73 ± 0.03 ^b^
Non-photochemical quenching (NPQ)	0.54 ± 0.02 ^bc^	0.49 ± 0.03 ^c^	0.86 ± 0.03 ^a^	0.62 ± 0.03 ^b^
Electron transport rate (ETR)	154 ± 6.2 ^a^	167 ± 5.1 ^a^	116 ± 4.2 ^c^	135 ± 5.3 ^b^
Plant dry mass (g plant^−1^)	6.11 ± 0.6 ^ab^	7.54 ± 0.4 ^a^	3.58 ± 0.4 ^c^	5.2 ± 0.5 ^b^
Leaf area (cm^2^ plant^−1^)	140 ± 5.7 ^b^	178 ± 6.1 ^a^	97.5 ± 3.8 ^d^	121 ± 4.6 ^c^

**Table 2 plants-08-00540-t002:** Electrolyte leakage, thiobarbituric acid reactive substances (TBARS) content, methylglyoxal (MG) content, lipoxygenase (LOX) activity in the leaf of mustard (*Brassica juncea* L.) cv. Varuna at 30 DAS. Plants were grown with/without zinc stress and treated with 200 µL L^−1^ ethephon at 20 DAS. Data are presented as treatment mean ± SE (*n* = 3). Data followed by same letter are not significantly different by LSD test at *p* < 0.05.

	Control	Ethephon	Zinc	Ethephon + Zinc
Electrolyte leakage (%)	1.24 ± 0.07 ^c^	0.81 ± 0.09 ^d^	3.93 ± 0.34 ^a^	2.08 ± 0.15 ^b^
TBARS content (nmol g^−^^1^ FW)	13.24 ± 0.6 ^c^	8.23 ± 0.4 ^d^	25.84 ± 0.8 ^a^	18.56 ± 0.7 ^b^
Methylglyoxal content (mol g^−1^ FW)	45.7 ± 1.7 ^d^	55.9 ± 2.1 ^c^	73.7 ± 3.0 ^a^	64.8 ± 2.1 ^b^
Lipoxygenase activity (µmol min^−1^ mg^−1^ protein)	10.2 ± 0.6 ^c^	7.9 ± 0.6 ^d^	15.4 ± 0.6 ^a^	13.5 ± 0.6 ^b^

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
