# Peer review of "Exogenously-Sourced Ethylene Modulates Defense Mechanisms and Promotes Tolerance to Zinc Stress in Mustard (Brassica juncea L.)"

_plants, 2019, doi:10.3390/plants8120540_

Round 1
Reviewer 1 Report
This paper describes the positive effect of ethylene on response of mustard seedlings to Zinc toxicity. The authors have made an impressive number of experiments, mostly at the biochemical level. From their data they propose that ethylene is able to mitigate the negative effect of Zinc on growth and photosynthesis, mostly through a stimulation of antioxidant mechanisms which can prevent oxidative damage. The experimental design is appropriate to address the question, the conclusions are quite clear (but largely uncomplete, see below) but not always supported by experimental data (see below).
I have some comments that should be addressed:
1/ I suggest to reorganize the MS which is not built in logical way. This would be easier for the reader if the effects of Zinc on growth and photosynthesis would be presented first (Figure 7 and Table 2). This is important to describe the effect of Zinc toxicity before showing the possible mechanisms associated with it. This will also imply to rebuild the organization of the discussion.
2/ I think that some facts have to be discussed more clearly in the discussion, which requires deep re-writing. The authors propose that Zn causes deregulation of photosynthetic apparatus via oxidative damage (lines 440-442). This is not properly explained. Please document about this mechanism.
3/ With regards to the data, it appear that PSII is not the only target of Zinc toxicity. See for example TBARS and electrolyte leakage measurements (Table 1): they suggest that Zinc also alters membrane properties. This is not at all evoked in the discussion. What is known about the effects of HM on biological membranes in the literature ?
4/ Antioxidant enzymes have different subcellular localizations. This is important to consider that point with regards to the proposed hypothesis (protection of PS II). Indeed this is not because an antioxidant enzyme will be activated that it will protect PSII, for example. What enzymes/systems are likely to be involved in chloroplast detoxication ? Could the other systems prevent damage to other cell compartments ? Please discuss.
5/ Some data are presented twice: see Figure 8. I suggest to remove this Figure and to suppress the data related to AVG effect. They don’t bring an added value to the MS, are incomplete (the effect of AVG is studied only for TBARS, dry mass and photosynthesis which is clearly not sufficient to draw conclusions) and poorly discussed in discussion (lines 459-462)
6/ The effect of ethylene on growth, photosynthesis or ROS scavenging and stimulation in plants is not described. Please document.
7/ The last paragraph (5. Conclusion and prospects) is not very pertinent. It is mostly a repetition of discussion and abstract and it does not propose clear perspectives of research. To be improved or suppressed.
The language must be improved, especially in the discussion.
Author Response
Reviewer 1
This paper describes the positive effect of ethylene on response of mustard seedlings to Zinc toxicity. The authors have made an impressive number of experiments, mostly at the biochemical level. From their data they propose that ethylene is able to mitigate the negative effect of Zinc on growth and photosynthesis, mostly through a stimulation of antioxidant mechanisms which can prevent oxidative damage. The experimental design is appropriate to address the question, the conclusions are quite clear (but largely uncomplete, see below) but not always supported by experimental data (see below).
Authors Response (AR): Authors thank Reviewer for appreciating the work
I suggest to reorganize the MS which is not built in logical way. This would be easier for the reader if the effects of Zinc on growth and photosynthesis would be presented first (Figure 7 and Table 2). This is important to describe the effect of Zinc toxicity before showing the possible mechanisms associated with it. This will also imply to rebuild the organization of the discussion.
AR: This has been done. The suggestion has improved our manuscript structure. As per suggestion, table of photosynthesis and growth has been modified accordingly.
I think that some facts have to be discussed more clearly in the discussion, which requires deep re-writing. The authors propose that Zn causes deregulation of photosynthetic apparatus via oxidative damage (lines 440-442). This is not properly explained. Please document about this mechanism.
AR: We have explained this point at proper place in the revised version.
With regards to the data, it appears that PSII is not the only target of Zinc toxicity. See for example TBARS and electrolyte leakage measurements (Table 1): they suggest that Zinc also alters membrane properties. This is not at all evoked in the discussion. What is known about the effects of HM on biological membranes in the literature?
AR: We have explained this point at proper place in discussion.
Antioxidant enzymes have different subcellular localizations. This is important to consider that point with regards to the proposed hypothesis (protection of PS II). Indeed this is not because an antioxidant enzyme will be activated that it will protect PSII, for example. What enzymes/systems are likely to be involved in chloroplast detoxication? Could the other systems prevent damage to other cell compartments? Please discuss.
AR: We have explained this point in discussion.
Some data are presented twice: see Figure 8. I suggest to remove this Figure and to suppress the data related to AVG effect. They don’t bring an added value to the MS, are incomplete (the effect of AVG is studied only for TBARS, dry mass and photosynthesis which is clearly not sufficient to draw conclusions) and poorly discussed in discussion (lines 459-462)
AR: AVG data have been removed to avoid repetitions
The effect of ethylene on growth, photosynthesis or ROS scavenging and stimulation in plants is not described. Please document.
AR: It has been included.
The last paragraph (5. Conclusion and prospects) is not very pertinent. It is mostly a repetition of discussion and abstract and it does not propose clear perspectives of research. To be improved or suppressed.
AR: Conclusions were re-written to avoid repetition
The language must be improved, especially in the discussion.
AR: We have thoroughly checked the MS for language correction.

Reviewer 2 Report
The manuscript has been well written.
Abstract – please explain high level of Zn in the soils ? According standard in Europe 200 mg Zn is normal level.
Introduction
Please give minimum 1 enzym, wchich response for physiological processes ......
I suggest add paper on assesment Ni on species plants:
For example:
Antonkiewicz J., Jasiewicz C., Koncewicz-Baran M., Sendor R. 2016. Nickel bioaccumulation by the chosen plant species. Acta Physiologiae Plantarum, 38, 40, pp.11. DOI: https://doi.org/10.1007/s11738-016-2062-5
Hypothesis and aim well written.
Material and methods
Were in analyzes using pure reagents ?
Was biological material enough for analysis?
In Science is several new method to determined N and P – Why used You old method ?
Results
Please justify (explain) the choice of plant for the experiment!
Please give effects used Zinc to the soil ..... –line 294
Line 341 - Why, because Zn precipitates nutrients ? Please explain this issue
Line 351 - Schuld be increase uptake of zinc from soil by plants ?
Discussion
In the chapter Discussion please provide resistant species of plants for Zn ....
Line 442 - Can be add recently paper about soil composition was the primary factor affecting chlorophyll fluorescence (CF) parameters.
For example:
Bączek-Kwinta R., Antonkiewicz J., Łopata-Stasiak A., Kępka W. 2019. Smoke compounds aggravate stress inflicted on Brassica seedlings by unfavourable soil conditions. Photosynthetica, 57, 1, 1-8. DOI: 10.32615/ps.2019.026
Conclusion
Correct, but please provide practis used ethylene to crop, to remediation of soils contamination of Zn ?
Author Response
Reviewer 2
The manuscript has been well written.
AR: Thanks for appreciation.
Abstract – please explain high level of Zn in the soils ? According standard in Europe 200 mg Zn is normal level.
AR: In Indian subcontinent 200 mg kg-1 Zn has been found toxic for plants. We have reported this in one of our manuscript published.
Introduction
Please give minimum 1 enzym, wchich response for physiological processes ......
AR: We have incorporated enzymes throughout the manuscript
I suggest add paper on assesment Ni on species plants:
For example:Antonkiewicz J., Jasiewicz C., Koncewicz-Baran M., Sendor R. 2016. Nickel bioaccumulation by the chosen plant species. ActaPhysiologiaePlantarum, 38, 40, pp.11. DOI: https://doi.org/10.1007/s11738-016-2062-5
AR: The suggested manuscript has been cited.
Hypothesis and aim well written.
AR: Thanks for the appreciation.
Material and methods
Were in analyzes using pure reagents?
AR: yes
Was biological material enough for analysis?
AR: Yes
In Science is several new method to determined N and P – Why used You old method ?
AR: Although several new methods are there for N and P analysis, but our protocol is one of the standard protocol and have been accepted in many of our manuscripts
Results
Please justify (explain) the choice of plant for the experiment!
AR: It is a well-known plant species of great scientific, economic and agronomic importance and used as a plant for scientific research as well as widely cultivated species. Moreover, our laboratory has around 20 years of experience on this crop.
Please give effects used Zinc to the soil ..... –line 294
AR: Done
Discussion
Line 442 - Can be add recently paper about soil composition was the primary factor affecting chlorophyll fluorescence (CF) parameters.
For example:Bączek-Kwinta R., Antonkiewicz J., Łopata-Stasiak A., Kępka W. 2019. Smoke compounds aggravate stress inflicted on Brassica seedlings by unfavourable soil conditions. Photosynthetica, 57, 1, 1-8. DOI: 10.32615/ps.2019.026
AR: We have cited this paper.
Conclusion
Correct, but please provide practis used ethylene to crop, to remediation of soils contamination of Zn?
AR: conclusion re-written.

Round 2
Reviewer 2 Report
The manuscript has been improved, add new references and improved method and discussion.
In the manuscript corrected and supplemented conclusions.